# Russian forests show strong potential for young forest growth

**Christopher S. R. Neigh** [1] ✉, **Paul M. Montesano** [1,2], **Joseph O. Sexton**[3], **Margaret Wooten**[1,4], **William Wagner**[1,4], **Min Feng**[3], **Nuno Carvalhais**[5], **Leonardo Calle**[1,6] & **Mark L. Carroll**[1]

Climate warming has improved conditions for boreal forest growth, yet the region's fate as a carbon sink of aboveground biomass remains uncertain. Forest height is a powerful predictor of aboveground forest biomass, and access to spatially detailed height-age relationships could improve the understanding of carbon dynamics in this ecosystem. The capacity of land to grow trees, defined in forestry as site index, was estimated by analyzing recent measurements of canopy height against a chronosequence of forest stand age derived from the historical satellite record. Forest-height estimates were then subtracted from the predicted site index to estimate height-age growth potential across the region. Russia, which comprised 73% of the forest change domain, had strong departures from model expectation of 2.4–4.8 ± 3.8 m for the 75th and 90th percentiles. Combining satellite observations revealed a large young forest growth sink if allowed to recover from disturbance.

Climate change has altered the ecophysiology of our planet, and the higher northern latitudes have been disproportionately affected by warming[1,2]. Numerous in-situ and remote-sensing studies have found that northern hemisphere vegetation could respond favorably to warming[3–11], but these effects could be transitory in space and time due to excess warming and drying observed in some regions[7,12–14]. Combined atmospheric and remote sensing studies have found high-latitude ecosystems are drawing progressively more carbon from the atmosphere[2,8], a phenomenon associated with arctic amplification. Many studies have identified the northern hemisphere to be a net carbon sink[15–17], while dendrochronology studies measuring incremental diameter wood growth suggest greater uncertainty in carbon flux dynamics[18–20].

The boreal biome is one of the largest on earth, accounting for 1/3rd of global forest area[1,21]. The region has experienced the most warming of any forest biome, annual surface temperatures have increased over 1.4 °C in the past century[22]. The boreal forest contains 38 ± 3.1 Pg of aboveground C[23], is underlain by 1672 Pg C — totaling 50% of global soil C, 88% of which is locked in perennially frozen soil[24], and sequesters ~20% of the total global forest C sink[25]. We include within our study all boreal forests/taiga ecoregions as well as tundra regions within boreal region boundaries[26].

Global warming is already affecting vegetation productivity[27], phenology[28], and C sequestration[8] across the northern high latitudes, and many other processes are impacting boreal forest health[1]. Forest structure varies dramatically across the high latitudes, even within the same species, due to local interactions between microclimate, topography, snow depth, wind, and edaphic conditions[29–31]. While earlier studies have extrapolated global maps of carbon accumulation by correlating in situ measurements with environmental covariates[32], these efforts have not been sufficiently comprehensive to infer growth potential across the entire boreal domain.

Boreal forests have long been considered to act as a sink of carbon, when excluding emissions from fossil fuel combustion, burning in wildfire, and outgassing from inland waters and wetlands[21]. Their mediation of greenhouse gas concentrations has amplified[8] and could reach a tipping point[33] wherein the accumulation of centuries old soil carbon[34] warms to a threshold that releases more $CO_2$ through respiration than is sequestered by photosynthesis in a given year[35]. The distribution of where and how the growth and/or decline of North American boreal forests are responding to climate has yet to be completely resolved due to the prior lack of widespread tree growth data[7,36,37]. The Eurasian taiga is even less certain due to poor access and the prevalence of larch (*Larix spp.*), a deciduous needle-leaf genus that could have a unique response to warming[38].

The majority of boreal above-ground biomass is found in Russia (63%[23]), its national role in the global forest carbon budget is only exceeded by the tropical forests in Brazil[39]. The Russian National Forest Inventory (NFI) reports no change in growing stock volume since the fall of the Soviet Union[40]. However, recent analysis with remote sensing found growing stock volume increases from 1988 to 2014 are equivalent to net growing stock

[1]NASA Goddard Space Flight Center, Greenbelt, MD, USA. [2]ADNET Systems Inc., Bethesda, MD, USA. [3]terraPulse Inc., North Potomac, MD, USA. [4]Science Systems Applications Inc., Lanham, MD, USA. [5]Max Planck Institute for Biogeochemistry, Jena, Germany. [6]University of Maryland, College Park, MD, USA. ✉e-mail: christopher.s.neigh@nasa.gov

volume losses in tropical countries[41]. Additionally, civil conflict has reduced access and data collection, creating a bias in our understanding of change in the Arctic-Boreal domain[42]. The global increase in biomass carbon stock has been found to be dominated by growth of young northern forests[27], but this could end this century[43]. Modest changes in climate could produce substantial and divergent impacts on growth and survival that would reorganize composition and structure[17,44].

Typically, satellite studies of vegetation growth in the northern hemisphere focus on the Normalized Difference Vegetation Index (NDVI), a unitless spectral index that estimates photosynthetic capacity of vegetation[45–47] that does not directly account for changes in vertical woody structure that satellite laser altimetry can provide. Boreal-wide estimates of height growth from spaceborne remote sensing at moderate spatial resolution (30 m) could resolve information about the status of these forests not considered with NDVI studies or reported by NFIs.

Site Index (SI) is a parameter widely used in forestry to describe the potential height-age growth of trees in a location or site[48]. Height growth is typically estimated from NFIs at the regional to country scale[49–52]. The concept of using remote sensing to estimate SI has been in the literature for decades[53], and a few studies have used airborne or spaceborne lidar with commercial very-high resolution stereo imagery to estimate forest height and used Landsat[54–59] or harvest data[60] to estimate forest age. However, no studies to our knowledge have attempted to estimate SI at continental scales, nor at the resolution necessary to depict the effect of disturbances on demographic patterns throughout the boreal domain.

Boreal SI can be difficult to estimate in situ due to the remoteness of much of the forest, thus many estimates of SI are found in southern, actively managed areas[49,50,61,62]. However, northern forests are well-suited to estimate SI with moderate (10–250 m) resolution remote sensing due to the patch size of stand clearing disturbances (tree cover falling below the minimum threshold in forest definition, i.e., 30%). Spatial information about the capacity of land to grow trees, coupled with subtracting estimates of heights of disturbed forest, could provide information about growth potential where young regenerating stands could sequester atmospheric carbon.

## Results

### Boreal forest height-age variance by country

The satellite record provides unprecedented spatial information on boreal forest height and stand age where coincident height-age pairs could be sampled. We provide an oblique-view example of combined height-age satellite data that captures some of the details these data can provide (Fig. 1). Over the past three decades, Western Eurasia has the oldest concentration of forest stands as compared to the rest of the biome (Fig. 2A). A majority of our sampled domain lies in Russia (58%) followed by Canada (29%), Alaska (6%), Finland (3%), Sweden (3%) and Norway (1%). The median age and standard deviation ($\pm$) of disturbed forest in Russia is $15 \pm 7.4^{-yrs}$ and $13 \pm 7.9^{-yrs}$ in Canada. Some of the oldest stands were found in Finland ($23 \pm 4.5^{-yrs}$), followed by Sweden ($21 \pm 6.2^{-yrs}$), Norway ($17.5 \pm 8.8^{-yrs}$) and Alaska ($16 \pm 3.8^{-yrs}$) (Supplementary Fig. 1A). Forests of western Eurasia provide a distributed height-age sample for growth curves (Supplementary Figs. 1B, C, 2–3) for that portion of the biome, while the remaining portions required aggregating height-age pairs and upscaling to 0.5° by 0.5° (5000-ha resolution) to provide enough samples to calculate SI and growth potential (Supplementary Fig. 4).

The tallest disturbed boreal stands are distributed throughout Eurasia (Fig. 2B). Russia has a median disturbed stand height of 16.5 ± 4.6 m, followed by Finland (16.4 ± 3 m), Sweden (14.6 ± 2.7 m) and Norway (13.7 ± 3.5 m). North America has disturbed boreal forest height of 9.9 ± 4.1 m in Canada and 9.4 ± 3.8 m in Alaska. Similarly, the tallest undisturbed boreal stands during the 1984 to 2020 Landsat record are concentrated in Eurasia (Fig. 2C). These stands could be 36 or more years old; this unknown is a well-known limitation of any Landsat-based time-series analysis. Russia has a median undisturbed height of 19.6 ± 5.9 m, followed by Finland (19.1 ± 4.1 m), Sweden (17.0 ± 4 m) and Norway (15.1 ± 4.1 m). North American undisturbed stand heights are less than Eurasia in Canada (13.4 ± 6 m) and Alaska (12.3 ± 3.8 m). Disturbance at this scale was greatest in Scandinavia and Western Europe, where more productive stands are harvested in small patches. Generally, human disturbance is more frequent in the southern boreal due to higher productivity and ease of access with harvest management, whereas wildfires are larger and more frequent in higher latitudes[1,21].

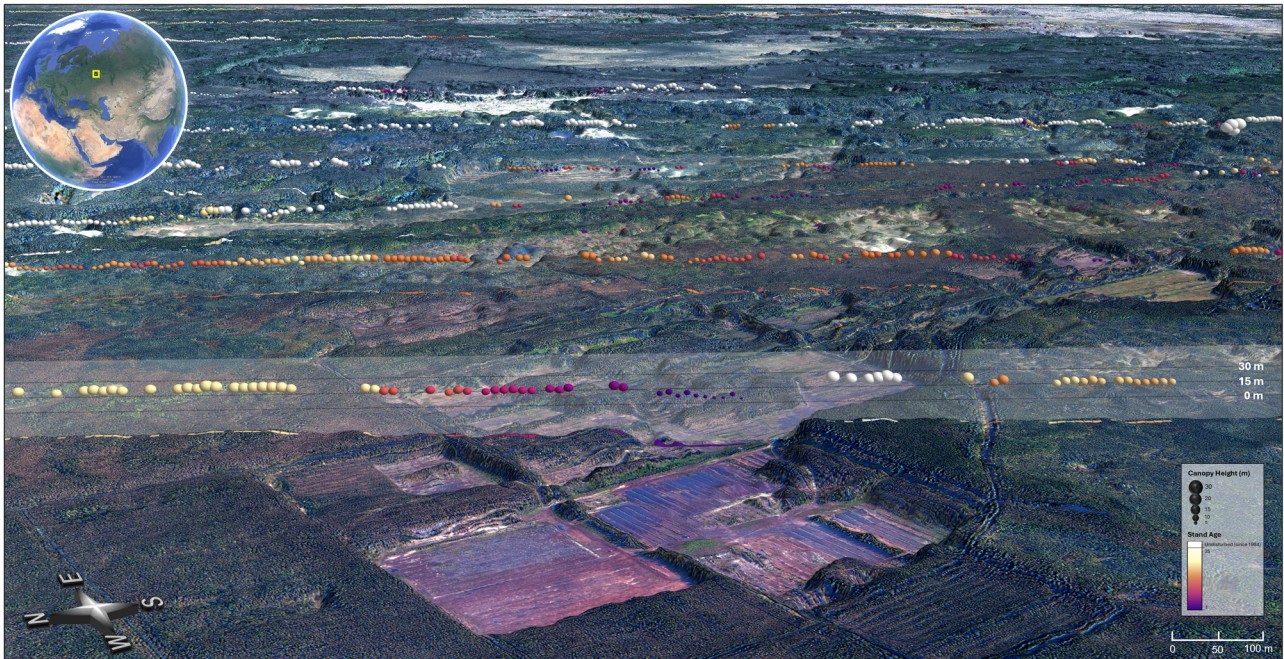

**Fig. 1 | Example of forest height observations from satellite data across a southern boreal landscape.** This site is located just outside of the Kerzhenskiy Gosudarstvennyy Nature preserve, 450 km east of Moscow centered on 56.5037° N, 44.6998° E. 2021 ICESat-2 tracks are displayed over a 2018 WorldView-1 Digital Surface Model. 98th percentile relative canopy heights (h_can_20m) from ICESat-2's ATL08 20 m vegetation product are depicted by height and sphere size, while associated stand ages from the Landsat-derived disturbance dataset are shown with a color gradient. ©Maxar 2018.

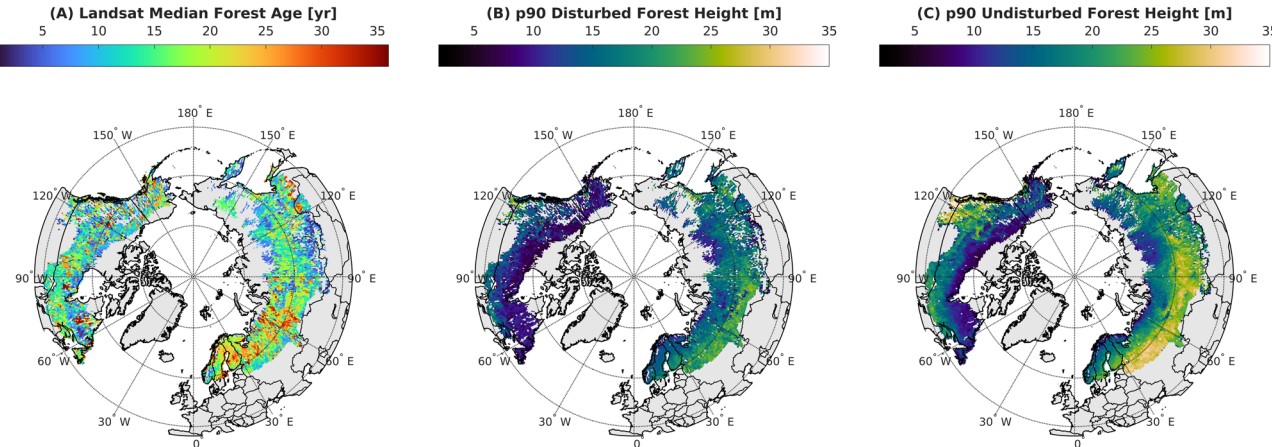

**Fig. 2 | Boreal forest height-age at 0.5° × 0.5°. A** Landsat median age, **B** 90th percentile of disturbed forest height, and **C** 90th percentile of undisturbed forest height from spatially coincident ICESat-2.

Portions of the boreal region with extensive fires typically have longer disturbance intervals and older forests than the satellite record. This can be observed as sampling intensity ($N < 500$ samples) in older forests, limiting the certainty of SI in these regions. Differences in disturbance type were not addressed in this analysis, yet the ability to resolve SI is associated with the frequency, size, and severity of disturbance, which are in turn directly associated with the ability to resolve forest age as time since stand clearing disturbance and stand height with remote sensing (Supplementary Figs. 4–5).

### Circumboreal SI provides an empirical reference of past forest growth

We provide a sample of the growth curve results for a few select cells along a latitudinal gradient in Western Eurasia (Fig. 3), additional samples are available in the supplementary materials (Supplementary Figs. 6–15). This is one of many samples where height-age relationships can asymptote, and a latitudinal gradient of 10 degrees reveals SI differences greater than 10 m. This approach relies on non-linear forest growth models that best fit the height-age data. Important empirical information was derived from millions of spaceborne observations of forest height stratified over 36 years. Complexities of variance in growth rates associated with scale differences and response lags from climate change impacts to growth were not investigated[63]. When aggregated to a relatively coarse scale, these data provide information about where the land can accumulate biomass and where hotspots of potential forest vertical growth exist.

### Hotspots of forest growth potential revealed in southwestern Russia

Our results provide a spaceborne estimate of the capacity of land to grow trees for the entire boreal forest when prior NFI reporting has been constrained to regions or states. We found hotspots of growth are strongest in southwestern Russia where forest height potential is substantial (>25 m, Fig. 2C). The spatial extent of these hotspots occurred west of the Ural Mountains where the density of Landsat observations (>90) was consistent with a majority of the domain (72.6%). These hotspots are less concentrated and sporadically extend into far Eastern Siberia and appear less frequently in North America. We present our unbiased models of growth centered on zero (Fig. 4A), where negative values represent overestimates of growth relative to predictions and positive values represent underestimates of growth relative to predictions and signify height-growth gaps. We focused our analysis to represent disturbance hotspots on areas that have gaps (Fig. 4B) greater than the 75th percentile of $z$-scores. This hotspot approach revealed most of the change exists in Russia (73%) followed by Canada (13%), Finland (7%), Alaska (3%), Sweden (3%) and Norway (<1%) (Fig. 4C, Supplementary Fig. 16). Most of the hotspots of growth with the 75th, 90th percentiles and uncertainty reported as the standard deviation (±)

exist in Russia (2.4–4.8 ± 3.8 m) and Canada (2.2–4.5 ± 3.6 m). Alaska had the largest gap (2.3–5.3 ± 3.7) but only accounted for 3% of the hotspot area. Sweden had the largest gap (1.5–2.5 ± 2.4 m) in Scandinavia but accounted for a small fraction of the hotspot area (3%). Finland accounted for a larger fraction of the hotspot (7%), but the growth gap was small (0.8–1.9 ± 1.9 m) and Norway had negligible hotspot (<1%) when compared to the entire boreal domain. Russian forests hold the second highest amount of above ground biomass nationally and have some of the most productive land to grow tall boreal forest stands over a large domain. Collating these satellite data revealed biome wide vertical growth potential when regular field reporting is not practical or maintained.

## Discussion

### Spaceborne vegetation mapping advances reveal a prominent hotspot of growth

Our results incorporate estimates of height along with the trend analysis that support the growth results of previous studies. As such, we normalized time to estimate the capacity of land to grow trees. Co-locating forest height and age data in a chronosequence revealed the distribution and size of gaps throughout the boreal biome. These data provide an independent source to compare to NFIs, where we find hotspots of growth in Russia to have the largest forest growth gap.

Our previous SI analysis used airborne lidar and Landsat stand age data used in this study[64], and validated estimates of Landsat stand age with NFI plots in Western Canada[54]. We found the mean and standard deviation of differences (residuals) between the modeled relationship and Landsat stand age is 13.6 ± 5.4 years. This mean value provides an estimated establishment time (years between disturbance event time and the point when a pixel exceeds a 30% tree canopy cover threshold value). The standard deviation represents an estimated variation in the time required for a forest to establish after a disturbance in the sampled areas. The validation of Landsat stand age has a Western Canadian boreal bias, is constrained to the 36-year satellite record, and we acknowledge this is a limitation of this study. However, the largest growth gap exists in western Russia, where many Landsat images were available to estimate age. This region has high human impact (harvesting, agriculture and settlements), is primarily managed forest, lacks permafrost, and has some of the lowest mean annual fraction of burned area (1997–2014)[1].

### Limitations of these forest growth estimates and future considerations

There are several limitations to this study that should be considered for future studies using remote sensing to estimate forest growth rates. First, we did not consider patch variance and disturbance type due to the limited density of height observations, nor did we consider environmental factors to

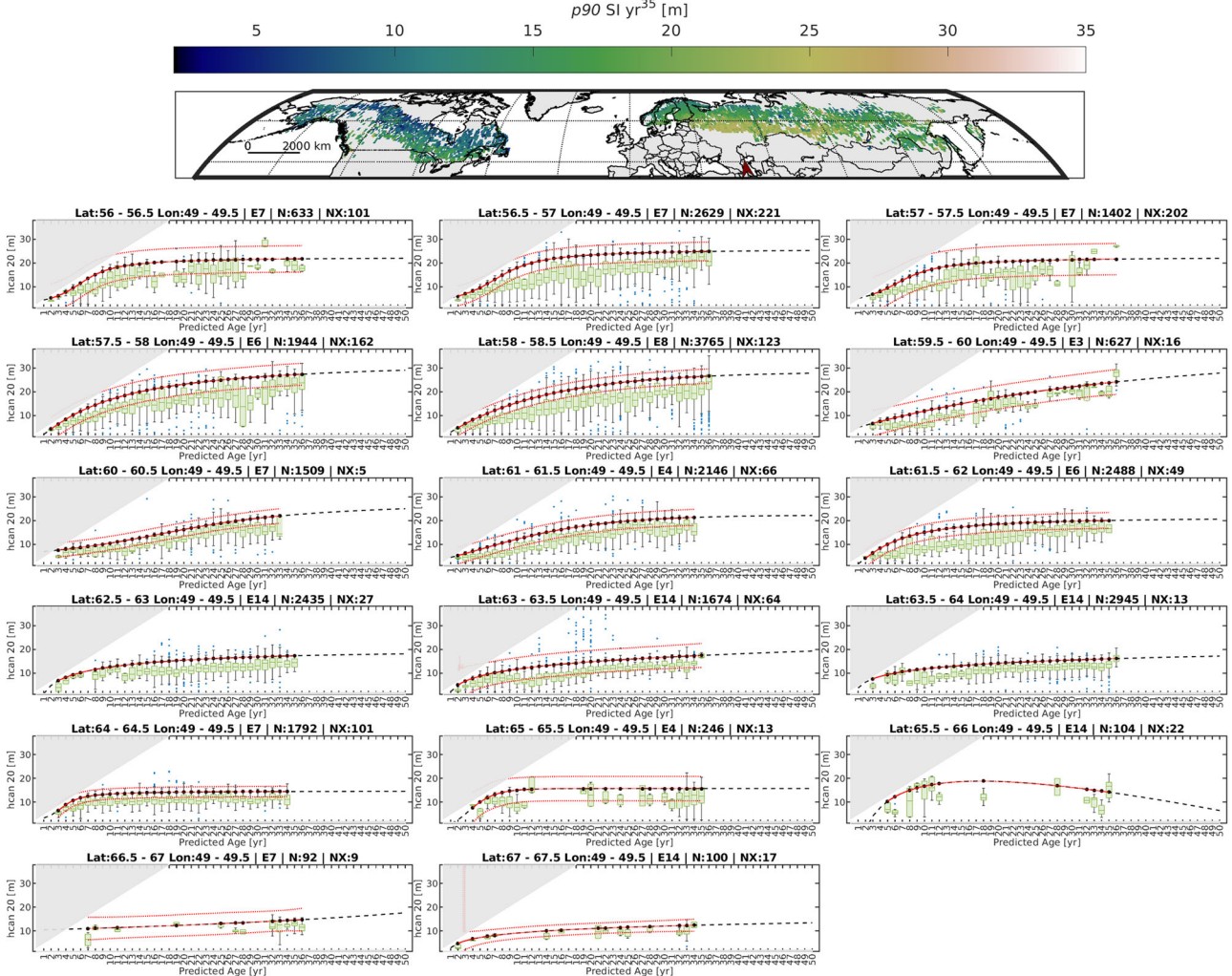

**Fig. 3 | Representative latitude transects of site index (SI).** The red north arrow indicates the location of the transect. Circumboreal SI derived from forest height-age relationship; rates presented are estimated on the 90th percentile of the expected height at the reference age (year 50), sequenced from south to north latitude. Green boxes indicate the upper and lower quartiles, green lines within boxes indicate the median, and red points indicate uncertainty bounds. Each chart title indicates the 0.5° × 0.5° area sampled with the equation (E) applied from Supplementary Table 1. Grayed area represents the samples excluded from analysis (NX), with the total number (N) of height-age pairs used displayed in chart titles.

predict growth where spatial gaps in this analysis currently exist. Second, spatial and temporal gaps in our data arise from sample limitations. Machine learning and/or deep learning with stereo imagery, SAR and LiDAR to fill gaps found in this analysis could be employed and/or approaches developed to fuse these datasets. Future studies would have more Landsat age samples and Ice Cloud and land Elevation Satellite 2 (ICESat-2) height observations that could be stratified by disturbance and or forest type. Landsat products for Canada demonstrate this capability[65,66], yet a circumboreal product is still needed. Improved data density and number of samples could also increase the ability to resolve SI at finer scales. Vegetation height products with higher sampling density throughout the boreal domain could be used in a similar growth curve approach and reveal patterns of growth rates that would be relevant to land managers and necessary for evaluating subtle shifts in aboveground carbon sequestration that occur across vast and remote spatial extents.

**Spaceborne data fusion rigor will improve growth estimates**
The ability to measure forest growth by estimating vegetation height and age over large areas will increase as planned and committed satellite missions become operational and systems to integrate surface topography and vegetation data mature. Some global datasets are currently available

(EarthDEM, Tandem-X) and many are forthcoming, with SAR missions (NISAR, Biomass, IceEye, etc.) and stereo missions (CO3D, Maxar Legion, Pleiades Neo, PRISM-2, CartoSat 2a/b, etc.) that provide global coverage for vegetation height mapping. These data could be combined with climate, soil, permafrost, and other covariates to predict the land's capacity to grow trees. Our approach provides empirical estimates of SI in boreal forests that are currently experiencing rapid warming. Russia's forests have been found to under-report biomass change in their NFI, yet they have the highest growth potential and contain the most above ground carbon in the biome. Recent widespread disturbance here has resulted in young, short stands that have the potential to have the greatest height growth within the entire boreal domain. Here we provide locations, estimates and methods documenting growth gaps of boreal forest stands. Future studies could use this information as a reference for potential change. Our chronosequence approach provides empirical spatial information that previously did not exist, about regions with potential biomass accumulation that could offset anthropogenic carbon emissions. Collating satellite data allows forest height-age assessment in remote inaccessible areas and aids independent verification of NFIs. Young boreal forests have been found to be a globally significant carbon sink and our results provide empirical evidence of their current potential.

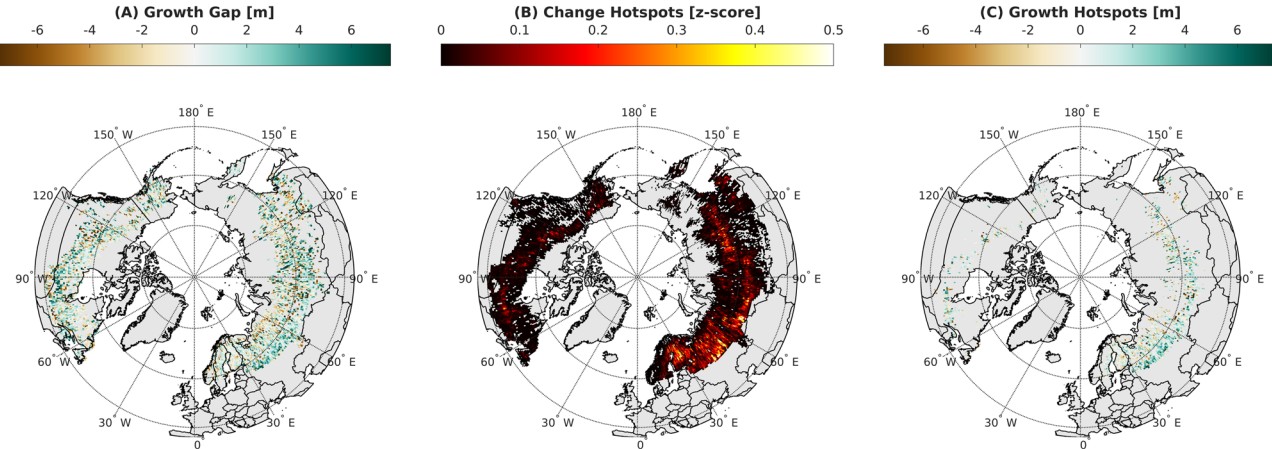

**Fig. 4 | Boreal forest growth and change hotspots. A** Boreal forest height-age growth potential calculated as expected minus observed forest height at the time of measurement. **B** Change hotspots calculated as the normalized *z*-score range of

segments disturbed throughout the domain. **C** Growth hotspots, calculated as the growth-gap that is greater than the third quartile of change hotspots.

## Materials and methods

### Estimating boreal forest growth rate

Forest height is correlated to woody biomass and carbon storage[67] and modeling height growth as a function of time enables prediction of above ground biomass potential. In this study, we had three distinct steps:

**1)** Build a database of spatially and temporally coincident forest height and age observations;

**2)** Map expected forest height across the region by fitting a range of forest growth models relating stand height to age and mapping the predictions across the boreal domain; and

**3)** Detect growth gaps by subtracting actual from expected forest height across the region.

This approach avoids more complex, mechanistic models for growth rate predictions[68–70]. We previously applied this approach and found vegetation growth in the continental United States using vegetation height from commercial very high-resolution stereo image pairs and time since disturbance from Landsat[55], and more recently in Canada and Alaska through a similar approach using Landsat age and airborne LiDAR from the Land Vegetation and Ice Sensor (LVIS-f) facility instrument[54].

### Estimating forest age and disturbance

Forest age was calculated based on 30-meter, annual-resolution estimates of tree cover spanning the boreal region from 1984 to 2020 derived from Landsat collection-1 surface reflectance images (http://landsat.usgs.gov). Leveraging the high degree of image overlap in the high latitudes, a total of 2189 Landsat Worldwide Reference System-2 (WRS-2) tiles were selected to provide complete coverage of the region. A maximum of four images were selected per year and WRS-2 tile to avoid noise from clouds and phenological variation. Images were combined from level-1 Terrain Corrected (L1T) Landsat 4 and 5 Thematic Mapper (TM), Landsat 7 Enhanced Thematic Mapper Plus (ETM + ), and Landsat 8 Operational Land Imager (OLI) sensors. Each image was converted to units of surface reflectance; the Landsat Ecosystem Disturbance Adaptive Processing System (LEDAPS)[71] was used for TM/ETM+ images, and the Landsat Surface Reflectance (LaSRC)[72] was used for OLI images. All images were scored by cloud coverage, seasonality, and image quality flags (e.g., SLC-off, Landsat collection 1 processing levels), and images with the highest scores in each year were selected for analysis:

$$score = ((1 - c) * (1 - w_s) + (s * w_s)) * w_q \quad (1)$$

where $c$ is the ratio of cloudiness in an image (0 = clear, 1 = fully cloudy); $s$ represents the seasonality of the image calculated as the number of days of an image acquisition to the mid-summer day:

$$s = \cos(abs(d - d_s) * 2/366) \quad (2)$$

where $d$ is the Julian day of the acquisition and $d_s$ is the value of Julian day of mid-summer; $w_s$ is a seasonality weight (higher in high latitudes and lower in low latitudes):

$$w_s = \sin(latitude) \quad (3)$$

and $w_q$ is an image quality weight, which is 0.1 for Landsat 7 ETM + SLC-off images collected after May 31, 2003, when the Scan Line Corrector (SLC) failed, and 1.0 for all other images.

Up to 148 images could be selected for a WRS-2 tile. Due to the actual availability of Landsat images, a total of 224,026 images were selected for analysis, including 110,407 TM images from Landsat 4 and 59,791 ETM+ images from Landsat 7, and 53,828 OLI images from Landsat 8. Of the 2189 WRS-2 tiles, 72.6% had at least 90 images collected out of the maximum possible 148, providing a sufficient sample for the analysis. Missed images for periods during the 1980s and 1990s occurred due to the lack of receiving capacities for Landsat 4 and 5; these tiles were mainly located in central and eastern Russia (Fig. S17). We provide an overview of our approach to estimate forest age and disturbance in supplementary Fig. S18.

Tree cover was estimated through a model $f$ of remotely sensed variables $X$ in any location $i$[73]:

$$\hat{c}_i = f\left(X; \hat{\beta}\right) + \varepsilon \quad (4)$$

where $c_i$ is the percentage of a pixel (i)'s area covered by woody vegetation taller than 3 to 5 meters; $\beta$ is a set of empirically estimated parameters; $\varepsilon$ is residual error or uncertainty; $X$ is a set of measurements of surface reflectance, derived indices, image acquisition date, and sensor identification. The model was fit to spatiotemporally coincident training data composed of 250-m, annual resolution estimates of tree cover between 2000 and 2019 as response and spectral measurements from coincident Landsat images as covariates and then applied to each complete Landsat image to produce the map of estimates. Following the United Nations Framework Convention on Climate Change (UNFCCC)[74], the category "forest" was defined as pixels exceeding a threshold of 30% tree cover[75], and the probability of a pixel belonging to forest was estimated as the integral of the normal probability

density function defined by the mean ($\hat{c}$) and variance ($\sigma$) of equation (1).

$$p(F)_{def} = p(c > c^\wedge *) = \int_{c*}^{100} p(c)dc \qquad (5)$$

where

$$p(c)_{def} = N(\hat{c}, \sigma^2) = (1/\sigma\sqrt{2\Pi})e^{-(c-\hat{c})^2/2\sigma^2} \qquad (6)$$

Pixels with ≤30 unobscured annual tree cover observations were excluded to minimize unbalanced representation caused by the lapses in the availability of images during the late 1980s and 1990s, mainly in central and northeast Siberia[76].

Forest losses and gains were detected in each pixel using a significance test of the forest-probabilities in two time periods, before and after each year as a moving window over years. A two-sample z-test was applied to the sample of forest-probabilities before (1) and after (2) each year in iteration:

$$z = \frac{\bar{x_1} - \bar{x_2}}{\sqrt{\frac{\sigma_1^2}{n_1} - \frac{\sigma_2^2}{n_2}}} \qquad (7)$$

where $\bar{x_1}$ and $\bar{x_2}$ are antecedent and trailing means, respectively, $\sigma_1$ and $\sigma_2$ are their standard deviations, and $n_1$ and $n_2$ are the number of forest-probability estimates contributing to the values in all years. The test was applied to each tree cover value through time with the kernel centered on 50% that was increasing over time - i.e., $p(F_{t_1}) = \bar{x_1} < 50\%$ and $p(F_{t_2}) = \bar{x_2} \geq 50\%$. If a statistically significant ($p <= 0.05$) difference was identified between the two ascending groups, the focal year was labeled as a gain or loss. Abrupt decreases below the 50% forest-probability threshold were labeled as forest loss (disturbance). The detected forest disturbance was categorized as "incomplete" if the annual tree cover had 7 years of missing records. Age was calculated by subtracting the year of the most recent significant gain from the focal year above 50% forest probability.

## Calculating change hotspots

We calculated z-scores to measure the age distance of the number of disturbed segments within a grid cell from the mean in terms of the standard deviation and rescaled the distance to be between 0 and 1. This metric was used to define domain wide change hotspots. We then limited z-score values to greater than the third quartile to define growth gap hotpots.

## Assembling vegetation heights

Stand-age data were combined with ICESat-2 forest height samples in 20 m × 11 along-track segments with non-linear forest growth models to estimate the lands capacity to grow trees and predict where vertical growth gaps exist. We assembled vegetation heights from the National Aeronautical and Space Administration's (NASA) Ice Cloud and Elevation Satellite-2 (ICESat-2) Advanced Topographic Laser Altimeter (ATLAS)-ATL08 height of canopy 20 m segments (hcan, 98% height profile) version 5 dataset[77] from the National Snow and Ice Data Center[78]. ATL08 data have been found to have strong agreement with National Terrestrial Ecosystem Monitoring System data in Canada with 100 m along-track segments[79]. Higher resolution along-track ATL08 segments have been found to have accuracies similar to those of the course (100 m × 11 m) segments in the boreal forest[80]. Vegetation heights from the ATL08 dataset were derived from along-track height differences (above the WGS84 ellipsoid) between ground and canopy surface elevations from each 20 m along-track segment. These height observations, h_canopy_20 m, represent the relative 98th percentile of height estimates from classified canopy photons for along-track segments for which there are at least 10 signal photons that include at least 3 canopy photons. These observations were gathered from granules acquired June to September, from 2019 to 2021 at circumpolar latitudes from 45° to 75° North. The segments were quality-filtered to include observations associated with strong beams; night and low sun angles (solar

elevation angle <5°); snow-free land surface; cloud-free; a valid 98th percentile height (h_can_20 m); terrain height difference from reference elevation <25 m; a total vertical geolocation error due to ranging and local slope <2.5 m; and land-cover based canopy height thresholds.

## Collating growth rates

We estimated growth rate patterns by collating coincident measurements of forest height and age. The locations of the point-based h_canopy_20 m observations, representing the 20 m × 11 m ATL08 segment centroids, provided the spatial index for extracting coincident stand age values. All quality filtered ATL08 observations corresponding with forested pixels were retained, resulting in 45,347,339 segments across the boreal domain. 39,259,745 forest age observations were older than the 36-year Landsat record, and 6,087,594 segments with coincident age estimates were available for analysis.

## Fitting forest growth models

We estimate SI by fitting 15 forest-growth models (Supplement Table 1) representing relationships between tree height and age[61,81–84]. A balance between available samples per grid cell and number of samples per year was needed to establish an adequate number of samples to predict forest growth. This approach required minimizing outliers for selecting growth curve parameters. We also excluded segments that had height estimates <2 and >40 m, and those that were collected on slopes >10°. A minimum of 30 segments per year and >5 non-continuous years were used to fit growth curves. Additionally, we applied a growth filter to determine valid heights with a minimum growth rate of >0.01 m per year above 1.4 m and below 3 m within the first year, and a maximum growth rate of 2 m per year. The range of this filter was broad to minimize the impact of outliers. Models were fitted and applied from height and age data within 5619 0.5° × 0.5° tiles to account for broad-scale spatial heterogeneity. We tested other gridding scales (0.1°, 0.25°, and 1°) and found 0.5° to have the optimal size to maximize the number of samples through time and resolve spatial growth gradients throughout the domain. Growth models were selected based on their use in chronosequence field plot estimates of non-linear growth of forests over much longer time intervals than what is available within the Landsat stand age record. While these curves are typically applied to height-dominant species within a plot, our approach considered the tallest trees within a segment with no consideration of differences in species or site conditions. Hence, this approach adopts standard forestry chronosequence approaches to estimate SI but is not directly analogous to existing field plot surveys due to the mixed sampling and the large domain of aggregation. However, millions of samples that are difficult and potentially inaccessible in portions of the boreal forest can be estimated where forest plots do not exist. Our area of analysis is a few orders of magnitude greater than what can be achieved with existing field plots and provides biome wide estimates.

## Statistical analysis for selecting a growth model

We selected 1 of 15 forest growth models per tile using root mean square errors (RMSEs) and sum squared errors (SSEs) to evaluate model performance. Models with RMSEs > 6, SSEs > 250, and curves predicting height in year 50 (i.e., $SI_{50}$) 50 < 2 or >40 m were excluded from analysis. Of the 15 models applied, those with the lowest RMSE and SSE were selected as output.

## Reporting summary

Further information on research design is available in the Nature Portfolio Reporting Summary linked to this article.

## Data availability

All data and materials used in the analyses are available on Github. https://github.com/mwooten3/ZonalStats-3DSI/tree/main/data.

## Code availability
All data and materials used in the analyses are available on Github. https://github.com/mwooten3/ZonalStats-3DSI/tree/main/data.

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

## Acknowledgements

We would like to thank the National Snow and Ice Data Center (NSIDC) for providing the ICESat-2 data and NASA Goddard's NCCS team for providing cloud-computing resources. This research was funded under grants from the National Aeronautics and Space Administration NNH16ZDA001N-CARBON (C.N., P.M., M.W., W.W., J.S., M.F., and L.C.), NNH21ZDA001N-TE (C.N. and P.M.) and NNH21ZDA001N-DSI (C.N. and P.M.). The use of trade names is intended for clarity only and does not constitute an endorsement of any product or company by the federal government.

## Author contributions

This study was designed by: C.N., P.M., J.S., M.W., and N.C. The manuscript was written by: C.N., M.W., P.M., and J.S. Manuscript reviews were performed by: C.N., M.W., P.M., J.S., W.W., L.C., M.C., N.C. and M.F. Funding for this work was provided by: C.N., M.C.

## Competing interests

The authors declare no competing interests.
