## [Transparent Peer Review file · Communications Earth & Environment]

Russian forests show strong potential for young forest growth

Corresponding Author: Dr Christopher Neigh

Version 0:

Decision Letter:

Dear Dr Neigh,

Your manuscript titled "Young forest growth potential revealed in Russia with circumboreal satellite analysis" has now been seen by 3 reviewers, and we include their comments at the end of this message. They find your work of interest, but some important points are raised. We are interested in the possibility of publishing your study in Communications Earth & Environment, but would like to consider your responses to these concerns and assess a revised manuscript before we make a final decision on publication.

In particular, we require that you:

- fully justify the use of LandSAT data, considering data limitations pre-2000
- adapt or fully justify your calculation approaches for forest age.

We therefore invite you to revise and resubmit your manuscript, along with a point-by-point response that takes into account the points raised. Please highlight all changes in the manuscript text file.

Please submit your point-by-point responses as a separate file, distinct from your cover letter where you can add responses to the Editors' comments that you do not want to be made available to the reviewers. Word files are preferred.

Important: The response to reviewers must not include any figures, tables or graphs. If you wish to respond to the reviewer reports with additional data in one of these formats, please add them to the main article or Supplementary Information, and refer to them in the rebuttal. Due to current technical limitations, any figures, tables, or graphs embedded in your rebuttal will not be included in the peer review file, if published.

Please use the following link to submit your revised manuscript, point-by-point response to the referees' comments (which should be in a separate document to any cover letter), a tracked-changes version of the manuscript (as a PDF file) and the completed checklist:

Link Redacted

We hope to receive your revised paper within six weeks; please let us know if you aren't able to submit it within this time so that we can discuss how best to proceed. If we don't hear from you, and the revision process takes significantly longer, we may close your file. In this event, we will still be happy to reconsider your paper at a later date, as long as nothing similar has been accepted for publication at Communications Earth & Environment or published elsewhere in the meantime.

Please do not hesitate to contact us if you have any questions or would like to discuss these revisions further. We look forward to seeing the revised manuscript and thank you for the opportunity to review your work.

Best regards,

Alice Drinkwater, PhD
Associate Editor
Communications Earth & Environment
@CommsEarth

EDITORIAL POLICIES AND FORMATTING

Editorial Policy: [Policy requirements](https://www.nature.com/documents/nr-editorial-policy-checklist.pdf) (Download the link to your computer as a PDF.)

- Behavioural and social science
- Ecological, evolutionary & environmental sciences
- Life sciences

<https://www.nature.com/documents/nr-reporting-summary.zip>

Furthermore, please align your manuscript with our format requirements, which are summarized on the following checklist: [Communications Earth & Environment formatting checklist](https://www.nature.com/documents/commsj-phys-style-formatting-checklist-article.pdf)

and also in our style and formatting guide [Communications Earth & Environment formatting guide](https://www.nature.com/documents/commsj-phys-style-formatting-guide-accept.pdf) .

*** DATA: Communications Earth & Environment endorses the principles of the Enabling FAIR data project (<http://www.copdess.org/enabling-fair-data-project/>). We ask authors to make the data that support their conclusions available in permanent, publically accessible data repositories. (Please contact the editor if you are unable to make your data available).

All Communications Earth & Environment manuscripts must include a section titled "Data Availability" at the end of the Methods section or main text (if no Methods). More information on this policy, is available at <http://www.nature.com/authors/policies/data/data-availability-statements-data-citations.pdf>.

If a community resource is unavailable, data can be submitted to generalist repositories such as [figshare](https://figshare.com/) or [Dryad Digital Repository](http://datadryad.org/). Please provide a unique identifier for the data (for example a DOI or a permanent URL) in the data availability statement, if possible. If the repository does not provide identifiers, we encourage authors to supply the search terms that will return the data. For data that have been obtained from publically available sources, please provide a URL and the specific data product name in the data availability statement. Data with a DOI should be further cited in the methods reference section.

REVIEWER COMMENTS:

Reviewer #1 (Remarks to the Author):

The authors estimated the capacity of land to grow trees (site index) by analyzing recent measurements of canopy height with a chronosequence of forest stand age derived from the historical satellite record. Forest-height estimates were then subtracted from the predicted site index to estimate height-age growth potential across the region. The authors believe that Russia accounts for 73% of the forested area of change and that young forest growth sinks could be significant if allowed to recover from disturbance. Although the topic is of importance, there are still some problems with the article. Major revision is needed.

1. Line 14. Adding to forest age.
2. Lines 40-41. annual surface temperatures increasing > 1.4 °C over the past century, which means the surface temperature increase of 1.4 °C per year, please explain.
3. Lines 46-47. How global warming is affecting vegetation productivity, phenology, and carbon in the northern high latitudes, please describe in detail and add relevant citations.
4. Line 93. What does stand clearing disturbances mean?
5. Line 103. What is sampling involved?
6. Figures. Why not present the results of the study using the polar coordinates, as compared to that presentation is better.
7. How to validate acquired tree age, SI?
8. Add the discussion on the uncertainties of forest age acquisition and the applications of SI in other study areas.
9. Lines 230-232. Please detail the rules for scoring Landsat imagery, as this determines what images will be selected to calculate tree cover.
10. Lines 269-271. The calculation of the age of the trees is not reasonable. If you want to estimate tree age from forest disturbance, you can use forest disturbance monitoring algorithms, such as LandTrendr, VCT, CCDC, etc., to obtain the year of the closest disturbance, and then subtract the year of the closest disturbance from the existing forest to obtain the age of the disturbed forest.
Shang, Rong, et al. "China's current forest age structure will lead to weakened carbon sinks in the near future." *The Innovation* 4.6 (2023).
Tian, Lei, et al. "Forest Age Mapping Using Landsat Time-Series Stacks Data Based on Forest Disturbance and Empirical Relationships between Age and Height." *Remote Sensing* 15.11 (2023): 2862.
11. Line 278, Assembling Forest (Tree) Heights.
12. Overall, the manuscript needs major revisions, and most importantly the authors need to improve on the data sources, tree age estimation, and discussion sections, otherwise I do not agree with publishing this manuscript.

Reviewer #2 (Remarks to the Author):

COMMSENV-24-1572-T Review report

In this manuscript, the authors analyze the large-scale circumboreal forest age-height relationships to discuss the forests' potential to sequester carbon. They present a new forest height growth modelling method based on the most recent remote-sensed estimations of forest age (ICESat-2) and age estimation based on Landsat-detected disturbance since 1985. Based on these models, they finally identify areas with significant gaps between potential forest height at fifty years (a metric largely used in forestry and a good proxy for above-ground biomass) and observed forest height. Such analyses are needed, given the important discussion about the potential of circumboreal forests to remain a carbon sink with future global change (combinations of changes in climate, fire, insect outbreaks, harvest, etc.). Overall, the analyses conducted were sound and well-explained, and the whole manuscript was well-written and pleasant to read. For all these reasons, I think this manuscript is worth publishing in the *Communications Earth and Environment*. I have only a few comments that I'll develop below.

My main concern is that I found it hard to understand if "undisturbed forests" were considered in the analysis. Landsat data can only detect disturbance (i.e. major changes in forest cover) since 1984, but then many "undisturbed forests" could also have been quite recently disturbed (e.g., < 50 years). I understand only recently disturbed forests (1984-2020) were used in the growth modelling analyses, which is fine, but I'm not sure how undisturbed forests were used to identify gaps between potential and observed height. The author should explain more clearly how forests with no disturbance detection since 1984 are considered in the different steps of the analyses.

I found some of the figure axes and legends hard to read.

Finally, authors may consider citing recent research on boreal forest growth:

D'Orangeville, L., D. Houle, L. Duchesne, R. P. Phillips, Y. Bergeron, and D. Kneeshaw. 2018. Beneficial effects of climate warming on boreal tree growth may be transitory. *Nature Communications* 9:3213.

Pau, M., S. Gauthier, R. D. Chavardès, M. P. Girardin, W. Marchand, and Y. Bergeron. 2021. Site index as a predictor of the effect of climate warming on boreal tree growth. *Global Change Biology*:gcb.16030.

Wang, J., A. Taylor, and L. D'Orangeville. 2022. Large, near-term increases in climate-induced tree growth may help offset impacts of increasing disturbance across Canadian boreal forest. *Research Square*.

Danneyyrolles, V., Y. Boucher, R. Fournier, and O. Valeria. 2023. Positive effects of projected climate change on post-disturbance forest regrowth rates in northeastern North American boreal forests. *Environmental Research Letters* 18:024041.

Reviewer #3 (Remarks to the Author):

This review pertains to the manuscript titled "Young Forest Growth Potential Revealed in Russia with Circumboreal Satellite Analysis." While I find this manuscript innovative and potentially impactful, I cannot recommend its acceptance in its current form for several main reasons.

First, the authors failed to address the extremely limited Landsat image availability during the pre-2000 era in Siberia, or they did not demonstrate how this limitation was managed. Whether this issue was actually addressed or not, it represents either a crucial flaw in the study design or a significant omission in the manuscript's reporting. Consequently, I have low confidence in the conclusions regarding Siberian forests, which the authors acknowledged in the introduction section are different from their North American counterparts.

Second, the manuscript includes several key terms, such as "Site Index (SI)" and "hotspots of growth," that are critical to the study's conclusions. However, the authors did not explicitly define these terms. Without such crucial information, it is nearly impossible to evaluate the scientific validity of the claims made.

Third, the discussion of the findings lacks depth. In its current form, the manuscript seems more like a demonstration of a continental-scale application of remote sensing coupled with machine learning techniques. Despite the apparent extensive work, the authors failed to emphasize the importance of their research. As a result, I find the discussion deeply unsatisfying, as it is rather brief and leaves much to be desired.

Below are my detailed comments:

Line 39: This sentence requires a citation.

Line 46 "vegetation productivity" needs to be followed by a citation.

Line 54: The first sentence needs a citation

Line 77 "does not account for changes in vertical woody structure": I don't agree with this statement. While it is true that NDVI has been found to be not sensible to forest structure in certain forest ecosystems, there are cases where positive relationships were found. Eg:

Gamon, J. A., Field, C. B., Goulden, M. L., Griffin, K. L., Hartley, A. E., Joel, G., ... & Valentini, R. (1995). Relationships between NDVI, canopy structure, and photosynthesis in three Californian vegetation types. *Ecological applications*, 5(1), 28-41.

Fiore, N. M., Goulden, M. L., Czimczik, C. I., Pedron, S. A., & Tayo, M. A. (2020). Do recent NDVI trends demonstrate boreal forest decline in Alaska?. *Environmental Research Letters*, 15(9), 095007.

Line 81: The details of SI were never given, not even in the method section

Line 91-93 "However, ... stand clearing disturbances": Is this statement from the literature or the authors? If the former, citations are needed. If the latter, there need to more information to back this up.

Line 145: I don't think how such "hotspots of growth" was identified was explained in the method section.

Line 223: It is well known that there were substantial data gaps in Landsat images' availability before 2000 over Siberia (Wulder et al 2016) which is why efforts looking into long-term forest dynamics and age distribution in Siberia using Landsat needed to take this into account (eg, Chen et al 2016). However, the authors failed to show how this data gap was accounted for over Siberian forests. Without this critical information, the authors can't convince me that the conclusion that was drawn is not biased towards the north american forests (even the authors acknowledged the potential differences between forests in siberia and those in North America in the introduction).

Wulder, M. A., White, J. C., Loveland, T. R., Woodcock, C. E., Belward, A. S., Cohen, W. B., ... & Roy, D. P. (2016). The global Landsat archive: Status, consolidation, and direction. *Remote Sensing of Environment*, 185, 271-283.

Chen, D., Loboda, T. V., Krylov, A., & Potapov, P. V. (2016). Mapping stand age dynamics of the Siberian larch forests from recent Landsat observations. *Remote sensing of environment*, 187, 320-331.

Line 282: The full name of Icesat-2 should be given the first time it shows up.

Communications Earth & Environment is committed to improving transparency in authorship. As part of our efforts in this direction, we are now requesting that all authors identified as 'corresponding author' create and link their Open Researcher and Contributor Identifier (ORCID) with their account on the Manuscript Tracking System prior to acceptance. ORCID helps the scientific community achieve unambiguous attribution of all scholarly contributions. You can create and link your ORCID from the home page of the Manuscript Tracking System by clicking on 'Modify my Springer Nature account' and following the instructions in the link below. Please also inform all co-authors that they can add their ORCIDs to their accounts and that they must do so prior to acceptance.

Version 1:

Decision Letter:

Dear Dr Neigh,

Your manuscript titled "Young forest growth potential revealed in Russia with circumboreal satellite analysis" has now been seen by our reviewers, whose comments appear below. In light of their advice we are delighted to say that we are happy, in principle, to publish a suitably revised version in Communications Earth & Environment.

We therefore invite you to revise your paper one last time to edit your manuscript to comply with our format requirements and to maximise the accessibility and therefore the impact of your work.

EDITORIAL REQUESTS:

****Please take care to match our formatting and policy requirements. We will check revised manuscript and return manuscripts that do not comply. Such requests will lead to delays. ****

SUBMISSION INFORMATION:

OPEN ACCESS:

Communications Earth & Environment is a fully open access journal. Articles are made freely accessible on publication. For further information about article processing charges, open access funding, and advice and support from Nature Research, please visit <https://www.nature.com/commsenv/open-access>

Link Redacted

** This url links to your confidential home page and associated information about manuscripts you may have submitted or be

reviewing for us. If you wish to forward this email to co-authors, please delete the link to your homepage first **

Best regards,

Alice Drinkwater, PhD
Associate Editor
Communications Earth & Environment
@CommsEarth

REVIEWERS' COMMENTS:

Reviewer #1 (Remarks to the Author):

The authors have carefully and thoroughly addressed all reviewer comments, and the revised manuscript is satisfactory. I recommend publication of the manuscripts in Communications Earth & Environment.

Reviewer #2 (Remarks to the Author):

The authors have successfully considered all my comments and suggestions. As such, I recommend the manuscript for publication.

REVIEWER COMMENTS:

Reviewer #1 (Remarks to the Author):

The authors estimated the capacity of land to grow trees (site index) by analyzing recent measurements of canopy height with a chronosequence of forest stand age derived from the historical satellite record. Forest-height estimates were then subtracted from the predicted site index to estimate height-age growth potential across the region. The authors believe that Russia accounts for 73% of the forested area of change and that young forest growth sinks could be significant if allowed to recover from disturbance. Although the topic is of importance, there are still some problems with the article. Major revision is needed.

Thank you for your insightful review of our manuscript. We have attempted to respond to all of your comments and we hope that our responses have addressed your concerns in the blue text below.

1. Line 14. Adding to forest age.

Added forest age to keywords.

2. Lines 40-41. annual surface temperatures increasing > 1.4 °C over the past century, which means the surface temperature increase of 1.4 °C per year, please explain.

Revised to clarify this statement is not a rate but a total amount. “The region has experienced the most warming of any forest biome, annual surface temperatures have increased over 1.4° C in the past century²¹.”

3. Lines 46-47. How global warming is affecting vegetation productivity, phenology, and carbon in the northern high latitudes, please describe in detail and add relevant citations.

Considering the manuscript is at the maximum word count and references, we provided a limited amount of details and 3 references that provide a broad context for warming impacts to the boreal forest. We selected these references because they are comprehensive, provide global context and are concise.

The prior text stated, “ (line 46) Global warming is already affecting vegetation productivity, phenology²⁶, and C (line 47) sequestration⁸ across the northern high latitudes, and many other processes are (line 48) impacting boreal forest health¹.”

1. Gauthier, S., Bernier, P., Kuuluvainen, T., Shvidenko, A. Z. & Schepaschenko, D. G. Boreal forest health and global change. *Science* **349**, 819–822 (2015).

8. Forkel, M. *et al.* Enhanced seasonal CO₂ exchange caused by amplified plant productivity in northern ecosystems. *Science* **351**, 696–699 (2016).

26. Forkel, M. *et al.* Identifying environmental controls on vegetation greenness phenology through model–data integration. *Biogeosciences* **11**, 7025–7050 (2014).

The Gauthier *et al.* citation provides an overview of boreal forest health, carbon content, fire dynamics, harvest and management of forest resources (Fig 1) and describes future climate projections under a range of scenarios that indicate a greater probability of boreal C stock decline (Fig 2). The first Forkel *et al.* citations provide global context with regard to satellite remote sensing (AVHRR, MODIS, NDVI) and ecosystem models (GPP, Temp, CO₂ amplitude, Fig 1) of how the high latitudes are experiencing large changes in vegetation and carbon cycle dynamics driven by climate-vegetation-carbon cycle feedbacks. The second Forkel reference describes how improved model data integration can improve seasonal to long-term vegetation greenness dynamics by including satellite observations of vegetation phenology into dynamic global vegetation models. We have added another more recent reference to describe northern hemisphere vegetation productivity increases to this section which we had cited later in the text.

40. Yang, H. *et al.* Global increase in biomass carbon stock dominated by growth of northern young forests over past decade. *Nat. Geosci.* 1–7 (2023) doi:10.1038/s41561-023-01274-4.

We hope our brevity in our text is addressed with these references.

4. Line 93. What does stand clearing disturbances mean?

Stand clearing disturbances are defined as the transition of a pixel from above to below 30% tree cover, following the UNFCCC⁷⁴ definition of “forest”. Using both the estimate of cover and its uncertainty, these are detected by a significance test of whether a pixel’s tree cover falls from above the 30% threshold of tree cover to below the threshold..(Lines 301-337).

5. Line 103. What is sampling involved?

Line 103 “Western Eurasia has the highest concentration of the oldest forest stands as compared to the rest of the biome (Figure 2A).” The sampling was forced to locations where age (Landsat 30 m pixels) height (ICESat-2 ATL08 filtered 20m geo segment) pairs are found. The median value per 0.5 degree pixel is reported for all of those samples.

We have revised the text to clarify this: “The satellite record provides unprecedented spatial information on boreal forest height and stand age where coincident height-age pairs could be sampled.”

Thank you for this suggestion.

6. Figures. Why not present the results of the study using the polar coordinates, as compared to that presentation is better.

We appreciate this comment and have changed most of the figures to a polar projection.

7. How to validate acquired tree age, SI?

We have validated our stand age product used in this study in another published manuscript using NFI data from Canada, and now added reference to that work in this manuscript.

Montesano, P. M. *et al.* Patterns of regional site index across a North American boreal forest gradient. *Environ. Res. Lett.* **18**, 075006 (2023).

Figure 5 in this reference shows the relationship (Landsat stand age = years-since-disturbance, n=120) and their distribution in North America.

Thank you for this comment.

8. Add the discussion on the uncertainties of forest age acquisition and the applications of SI in other study areas.

We have added the following text to the Discussion section: “Our previous site-index analysis used airborne lidar and Landsat stand age data used in this study⁶⁰, and validated estimates of Landsat stand age with NFI plots in Western Canada⁵¹. We found the mean and standard deviation of differences (residuals) between modeled relationship and Landsat stand age is 13.6 ± 5.4 years. This mean value provides an estimated establishment time (years between disturbance event time and the point when a pixel exceeds a 30% tree canopy cover threshold value). The standard deviation represents an estimated variation in the time required for a forest to establish after a disturbance in the sampled areas. The validation of Landsat stand age has a Western Canadian boreal bias, is constrained to the 36-year satellite record, and we acknowledge this is a limitation of this study.”

Thank you for this suggestion.

9. Lines 230-232. Please detail the rules for scoring Landsat imagery, as this determines what images will be selected to calculate tree cover.

We have added the following text and equations to clarify the scoring and selecting of Landsat images.

All images were scored by cloud coverage, seasonality, and image quality flags (e.g., SLC-off, Landsat collection 1 processing levels), and images with the highest scores in each year were selected for analysis:

$$\text{scoring index} = ((1 - c) * (1 - w_s) + (s * w_s)) * w_q,$$

where c is the ratio of cloudiness in an image (0 = clear, 1 = fully cloudy); s represents the seasonality of the image calculated as the number of days of an image acquisition to the mid-summer day:

$$s = \cos (\text{abs} (d - d_s) * 2 / 366),$$

where d is the Julian day of the acquisition and d_s is the value of Julian day of mid-summer; w_s is a seasonality weight (higher in high latitudes and lower in low latitudes):

$$w_s = \sin (\text{latitude}),$$

and w_q is an image quality weight, which is 0.1 for Landsat 7 ETM+ SLC-off images collected after May 31, 2003, when the Scan Line Corrector (SLC) failed, and 1.0 for all other images.

10. Lines 269-271. The calculation of the age of the trees is not reasonable. If you want to estimate tree age from forest disturbance, you can use forest disturbance monitoring algorithms, such as LandTrendr, VCT, CCDC, etc., to obtain the year of the closest disturbance, and then subtract the year of the closest disturbance from the existing forest to obtain the age of the disturbed forest.

Shang, Rong, et al. "China's current forest age structure will lead to weakened carbon sinks in the near future." *The Innovation* 4.6 (2023).

Tian, Lei, et al. "Forest Age Mapping Using Landsat Time-Series Stacks Data Based on Forest Disturbance and Empirical Relationships between Age and Height." *Remote Sensing* 15.11 (2023): 2862.

Thank you for this comment. We applied a well-established algorithm to estimate forest disturbance, an extension of the algorithm developed by our coauthors that was

responsible for producing the NASA Global Forest Cover and Change Dataset (Sexton et al. 2013, 2016a, b). This algorithm (which admittedly suffers from a lack of appealing branding compared to LandTrendr, VCT, etc.) is similar to the others in that it calculates a variable quantifying the likelihood of a pixel being forest, but it differs from these approaches by using estimates and associated per-pixel uncertainties (RMSE) of tree cover instead of the more abstract “Forestness Index” employed in VCT and others. The algorithm thus improves the statistical robustness of the estimates of cover and change.

References:

Kim, D.-H.; **Sexton, J. O.**; Noojipady, P.; Huang, C.; Anand, A.; Channan, S.; **Feng, M.**; Townshend, J. R. Global, Landsat-Based Forest-Cover Change from 1990 to 2000. *Remote Sensing of Environment* **2014**, *155*, 178–193.
<https://doi.org/10.1016/j.rse.2014.08.017>.

Sexton, J.O., Feng, M., S. Channan, X.-P. Song, D.-H. Kim, P. Noojipady, E.F. Vermote, D. Song, C. Huang, K. Collins, J.R.G. Townshend. 2016. NASA Earth Science Data Records of Global Forest Cover and Change. NASA Algorithm Theoretical Basis Document.

Sexton, J.O., P. Noojipady, A. Anand, X.-P. Song, C. Huang, S.M. McMahon, **M. Feng**, S. Channan, J.R. Townshend. 2015. A model for the propagation of uncertainty from continuous estimates of tree cover to categorical forest cover and change. *Remote Sensing of Environment* *156*: 418-425.

Sexton, J.O., X.-P. Song, **M. Feng**, P. Noojipady, A. Anand, C. Huang, D.-H. Kim, K.M. Collins, S. Channan, C. DiMiceli, J.R. Townshend. 2013. Global, 30-m resolution continuous fields of tree cover: Landsat-based rescaling of MODIS continuous fields and lidar-based estimates of error. *International Journal of Digital Earth* *6*(5): 427-448.

11. Line 278, Assembling Forest (Tree) Heights.

12. Overall, the manuscript needs major revisions, and most importantly the authors need to improve on the data sources, tree age estimation, and discussion sections, otherwise I do not agree with publishing this manuscript.

Thank you for your review of our manuscript. Considering the prior text was at the maximum word count for a manuscript we included as much detail as possible to address your comments and questions.

Reviewer #2 (Remarks to the Author):

COMMSENV-24-1572-T Review report

In this manuscript, the authors analyze the large-scale circumboreal forest age-height relationships to discuss the forests' potential to sequester carbon. They present a new forest height growth modelling method based on the most recent remote-sensed estimations of forest age (ICESat-2) and age estimation based on Landsat-detected disturbance since 1985. Based on these models, they finally identify areas with significant gaps between potential forest height at fifty years (a metric largely used in forestry and a good proxy for above-ground biomass) and observed forest height. Such analyses are needed, given the important discussion about the potential of circumboreal forests to remain a carbon sink with future global change (combinations of changes in climate, fire, insect outbreaks, harvest, etc.). Overall, the analyses conducted were sound and well-explained, and the whole manuscript was well-written and pleasant to read. For all these reasons, I think this manuscript is worth publishing in the Communications Earth and Environment. I have only a few comments that I'll develop below.

Thank you for these kind words and taking the time to review our manuscript.

My main concern is that I found it hard to understand if “undisturbed forests” were considered in the analysis. Landsat data can only detect disturbance (i.e. major changes in forest cover) since 1984, but then many “undisturbed forests” could also have been quite recently disturbed (e.g., < 50 years). I understand only recently disturbed forests (1984-2020) were used in the growth modelling analyses, which is fine, but I'm not sure how undisturbed forests were used to identify gaps between potential and observed height. The author should explain more clearly how forests with no disturbance detection since 1984 are considered in the different steps of the analyses.

Thank you for this comment and we have revised the text to address your concern. Based on concerns from the other reviewers we have added more details about how disturbed and undisturbed forests are calculated. We used co-located disturbed and undisturbed Landsat pixels and ICESat-2 geo segments as a reference of height older than the 1984-2020 Landsat record. Yes, these forests could be 36 years old or 300+ years old. We found including the undisturbed forest heights to be a sanity check on the SI analysis.

We now provide additional text for our recently published work on validating forest age estimates. We have validated our stand age product used in this study in another published manuscript using NFI data from Canada.

Montesano, P. M. *et al.* Patterns of regional site index across a North American boreal forest gradient. *Environ. Res. Lett.* **18**, 075006 (2023).

The gap analysis was performed with the space-for-time/age-height analysis to estimate the growth potential within a 0.5 x 0.5 gridcell. Potential (SI) then had disturbed heights subtracted to estimate gaps.

We have added text to the manuscript enhancing the description of Fig 2. C Undisturbed forest height: “Similarly, the tallest (90th height percentile) undisturbed boreal stands during the 1984 to 2020 Landsat record are concentrated in Eurasia (Figure 2C). These stands could be 36 or more years old, this unknown is a limitation of our approach.”

I found some of the figure axes and legends hard to read.

We have enlarged most of the figures to make them more legible.

Finally, authors may consider citing recent research on boreal forest growth:

Thank you for your suggested references. We have added some of these references to the manuscript.

D’Orangeville, L., D. Houle, L. Duchesne, R. P. Phillips, Y. Bergeron, and D. Kneeshaw. 2018. Beneficial effects of climate warming on boreal tree growth may be transitory. *Nature Communications* 9:3213.

Pau, M., S. Gauthier, R. D. Chavardès, M. P. Girardin, W. Marchand, and Y. Bergeron. 2021. Site index as a predictor of the effect of climate warming on boreal tree growth. *Global Change Biology*:gcb.16030.

Wang, J., A. Taylor, and L. D’Orangeville. 2022. Large, near-term increases in climate-induced tree growth may help offset impacts of increasing disturbance across Canadian boreal forest. *Research Square*.

Danneyrolles, V., Y. Boucher, R. Fournier, and O. Valeria. 2023. Positive effects of projected climate change on post-disturbance forest regrowth rates in northeastern North American boreal forests. *Environmental Research Letters* 18:024041.

Reviewer #3 (Remarks to the Author):

This review pertains to the manuscript titled “Young Forest Growth Potential Revealed in Russia with Circumboreal Satellite Analysis.” While I find this manuscript innovative and potentially impactful, I cannot recommend its acceptance in its current form for several main reasons.

First, the authors failed to address the extremely limited Landsat image availability during the pre-2000 era in Siberia, or they did not demonstrate how this limitation was managed. Whether this issue was actually addressed or not, it represents either a crucial flaw in the study design or a significant omission in the manuscript’s reporting. Consequently, I have low confidence in the conclusions regarding Siberian forests, which the authors acknowledged in the introduction section are different from their North American counterparts.

Thank you for your thorough review. We have added more text to the methods section based on comments from you and the other reviewers' concerns about the density of the Landsat record indicated in quotes below. We have also added a supplemental figure that shows the density of images used per WRS-2 tile.

Up to 148 images could be selected for a WRS-2 tile. A total of 224,026 images were selected for analysis, including 110,407 TM images from Landsat 4 and 59,791 ETM+ images from Landsat 7, and 53,828 OLI images from Landsat 8. “Of the 2,189 WRS-2 tiles, 72.6% had at least 90 images collected out of the maximum possible 148, providing a sufficient sample for the analysis. Missed images for periods during the 1980s and 1990s occurred due to the lack of receiving capacities for Landsat 4 and 5; these tiles were mainly located in central and eastern Russia (Fig. S17).”

We have also added text that clarifies and corrects this perceived crucial flaw in quotes below: Our results provide a spaceborne estimate of the capacity of land to grow trees for the entire boreal forest when prior NFI reporting has been constrained to regions or states. We found hotspots of growth are strongest in southwestern Russia where forest height potential is substantial (> 25 m, Figure 2C). “The spatial extent of the largest growth hotspots occurred west of the Ural Mountains where the density of Landsat observations (>90) were consistent with a majority of the domain (72.6%).”

We also added the following text to the discussion section to address the concern bias in data: “Our previous site-index analysis used airborne lidar and Landsat stand age data used in this study⁶⁰, and validated estimates of Landsat stand age with NFI plots in

Western Canada⁵¹. We found the mean and standard deviation of differences (residuals) between modeled relationship and Landsat stand age is 13.6 ± 5.4 years. This mean value provides an estimated establishment time (years between disturbance event time and the point when a pixel exceeds a 30% tree canopy cover threshold value). The standard deviation represents an estimated variation in the time required for a forest to establish after a disturbance in the sampled areas. The validation of Landsat stand age has a Western Canadian boreal bias, is constrained to the 36-year satellite record, and we acknowledge this is a limitation of this study. However, the largest growth gap exists in western Russia, where many Landsat images were available to estimate age.”

Second, the manuscript includes several key terms, such as “Site Index (SI)” and “hotspots of growth,” that are critical to the study's conclusions. However, the authors did not explicitly define these terms. Without such crucial information, it is nearly impossible to evaluate the scientific validity of the claims made.

We provided many of these details in the introduction and materials and methods sections.

Site index was defined in the introduction section throughout the last two paragraphs with 12 references. We now have added text in quotes below to the methods section, Fitting Forest Growth Models: We “estimate site index by” fitting 15 forest-growth models (Supplement Table 1) representing relationships between tree height and age^{77–81}.

The hotspot of growth was defined in the methods section, Calculating Change Hotspots. We calculated z-scores to measure the age distance of the number of disturbed segments within a grid cell from the mean in terms of the standard deviation and rescaled the distance to be between 0 and 1. This metric was used to define domain wide change hotspots. We then limited z-score values to greater than the third quartile to define growth gap hotspots.

Considering word count limitations of the journal and the importance of content in other sections of the manuscript we think the text is comprehensive.

Third, the discussion of the findings lacks depth. In its current form, the manuscript seems more like a demonstration of a continental-scale application of remote sensing coupled with machine learning techniques. Despite the apparent extensive work, the authors failed to emphasize the importance of their research. As a result, I find the discussion deeply unsatisfying, as it is rather brief and leaves much to be desired.

Thank you for this comment.

The primary section header labeled Discussion, was extremely brief and yes, we agree unsatisfying if you do not consider the following section. Our intent was the following section titled “Considerations for Future Boreal Forest Growth Rate Estimates” to be part of the discussion section. Nevertheless, we have added text to the primary labeled Discussion, to give the manuscript more depth. “Our previous site-index analysis used airborne lidar and Landsat stand age data used in this study⁶⁰, and validated estimates of Landsat stand age with NFI plots in Western Canada⁵¹. We found the mean and standard deviation of differences (residuals) between modeled relationship and Landsat stand age is 13.6 ± 5.4 years. This mean value provides an estimated establishment time (years between disturbance event time and the point when a pixel exceeds a 30% tree canopy cover threshold value). The standard deviation represents an estimated variation in the time required for a forest to establish after a disturbance in the sampled areas. The validation of Landsat stand age has a Western Canadian boreal bias, is constrained to the 36-year satellite record, and we acknowledge this is a limitation of this study. However, the largest growth gap exists in western Russia, where many Landsat images were available to estimate age. This region has high human impact (harvesting, agriculture and settlements), is primarily managed forest and lacks permafrost; and has some of the lowest mean annual fraction of burned area (1997-2014)¹.”

Below are my detailed comments:

Line 39: This sentence requires a citation.

Thank you for the suggestion, corrected.

Line 46 “vegetation productivity” needs to be followed by a citation.

Thank you for the suggestion, corrected.

Line 54: The first sentence needs a citation

Thank you for the suggestion, revised text, “Boreal forests have long been considered to act as a sink of carbon, when excluding emissions from fossil fuel combustion, burning in wildfire, and outgassing from inland waters and wetlands²¹”.

Line 77 “does not account for changes in vertical woody structure”: I don't agree with this statement. While it is true that NDVI has been found to be not sensible to forest structure in certain forest ecosystems, there are cases where positive relationships were found. Eg:

Gamon, J. A., Field, C. B., Goulden, M. L., Griffin, K. L., Hartley, A. E., Joel, G., ... & Valentini, R. (1995). Relationships between NDVI, canopy structure, and photosynthesis in three Californian vegetation types. *Ecological applications*, 5(1), 28-41.

Fiore, N. M., Goulden, M. L., Czimczik, C. I., Pedron, S. A., & Tayo, M. A. (2020). Do recent NDVI trends demonstrate boreal forest decline in Alaska?. *Environmental Research Letters*, 15(9), 095007.

Thank you for this suggestion, we added text in quotes: Typically, satellite studies of vegetation growth in the northern hemisphere focus on the Normalized Difference Vegetation Index (NDVI), a unitless spectral index that estimates photosynthetic capacity of vegetation^{44–46} that does not “directly” account for changes in vertical woody structure “that satellite laser altimetry can provide”.

Line 81: The details of SI were never given, not even in the method section

We disagree, we provided many of these details in the introduction and materials and methods sections.

Site index was defined in the introduction section throughout the last two paragraphs with 12 references. We now have added text in quotes below to the methods section, Fitting Forest Growth Models: We “estimate site index by” fitting 15 forest-growth models (Supplement Table 1) representing relationships between tree height and age^{77–81}.

Considering word count limitations of the journal and the importance of content in other sections of the manuscript we think the text is comprehensive.

Line 91-93 “However, ... stand clearing disturbances”: Is this statement from the literature or the authors? If the former, citations are needed. If the latter, there need to more information to back this up.

Thank you for this comment. We have revised the text and added references for context. “Boreal SI is difficult to estimate *in situ* due to the remoteness of much of the forest, thus many estimates of SI are found in southern, actively managed areas^{48,49,59,60}.”

Line 145: I don’t think how such “hotspots of growth” was identified was explained in the method section.

We provided these details in the materials and methods sections.

The hotspot of growth was defined in the methods section, Calculating Change Hotspots. We calculated z-scores to measure the age distance of the number of disturbed segments within a grid cell from the mean in terms of the standard deviation and rescaled the distance to be between 0 and 1. This metric was used to define domain wide change hotspots. We then limited z-score values to greater than the third quartile to define growth gap hotspots.

Considering word count limitations of the journal and the importance of content in other sections of the manuscript we think the text is comprehensive.

Line 223: It is well known that there were substantial data gaps in Landsat images' availability before 2000 over Siberia (Wulder et al 2016) which is why efforts looking into long-term forest dynamics and age distribution in Siberia using Landsat needed to take this into account (eg, Chen et al 2016). However, the authors failed to show how this data gap was accounted for over Siberian forests. Without this critical information, the authors can't convince me that the conclusion that was drawn is not biased towards the north american forests (even the authors acknowledged the potential differences between forests in siberia and those in North America in the introduction).

Wulder, M. A., White, J. C., Loveland, T. R., Woodcock, C. E., Belward, A. S., Cohen, W. B., ... & Roy, D. P. (2016). The global Landsat archive: Status, consolidation, and direction. *Remote Sensing of Environment*, 185, 271-283.

Chen, D., Loboda, T. V., Krylov, A., & Potapov, P. V. (2016). Mapping stand age dynamics of the Siberian larch forests from recent Landsat observations. *Remote sensing of environment*, 187, 320-331.

Thank you for this comment and this was mentioned by the other reviewers.

We have added more text to the methods section based on comments from you and the other reviewers' concerns about the density of the Landsat record indicated in quotes below. We have also added a supplemental figure that shows the density of images used per WRS-2 tile.

Up to 148 images could be selected for a WRS-2 tile. A total of 224,026 images were selected for analysis, including 110,407 TM images from Landsat 4 and 59,791 ETM+ images from Landsat 7, and 53,828 OLI images from Landsat 8. "Of the 2,189 WRS-2

tiles, 72.6% had at least 90 images collected out of the maximum possible 148, providing a sufficient sample for the analysis. Missed images for periods during the 1980s and 1990s occurred due to the lack of receiving capacities for Landsat 4 and 5; these tiles were mainly located in central and eastern Russia (Fig. S17).”

We have also added text that clarifies and corrects this perceived critical flaw in quotes below: Our results provide a spaceborne estimate of the capacity of land to grow trees for the entire boreal forest when prior NFI reporting has been constrained to regions or states. We found hotspots of growth are strongest in southwestern Russia where forest height potential is substantial (> 25 m, Figure 2C). “The spatial extent of the largest growth hotspots occurred west of the Ural Mountains where the density of Landsat observations (>90) were consistent with a majority of the domain (72.6%).”

We also added the following text to the discussion section to address the concern bias in data: “Our previous site-index analysis used airborne lidar and Landsat stand age data used in this study⁶⁰, and validated estimates of Landsat stand age with NFI plots in Western Canada⁵¹. We found the mean and standard deviation of differences (residuals) between modeled relationship and Landsat stand age is 13.6 ± 5.4 years. This mean value provides an estimated establishment time (years between disturbance event time and the point when a pixel exceeds a 30% tree canopy cover threshold value). The standard deviation represents an estimated variation in the time required for a forest to establish after a disturbance in the sampled areas. The validation of Landsat stand age has a Western Canadian boreal bias, is constrained to the 36-year satellite record, and we acknowledge this is a limitation of this study. However, the largest growth gap exists in western Russia, where many Landsat images were available to estimate age.”

Line 282: The full name of Icesat-2 should be given the first time it shows up.

Corrected, thank you.

Communications Earth & Environment is committed to improving transparency in authorship. As part of our efforts in this direction, we are now requesting that all authors identified as 'corresponding author' create and link their Open Researcher and Contributor Identifier (ORCID) with their account on the Manuscript Tracking System prior to acceptance. ORCID helps the scientific community achieve unambiguous attribution of all scholarly contributions. You can create and link your ORCID from the home page of the Manuscript Tracking System by clicking on 'Modify my Springer Nature account' and following the instructions in the link below. Please also inform all co-authors that they can add their ORCID to their accounts and that they must do so prior to acceptance.

If you experience problems in linking your ORCID, please contact the Platform Support Helpdesk.

This email has been sent through the Springer Nature Tracking System NY-610A-NPG&MTS

Confidentiality Statement:

This e-mail is confidential and subject to copyright. Any unauthorised use or disclosure of its contents is prohibited. If you have received this email in error please notify our Manuscript Tracking System Helpdesk team at <http://platformsupport.nature.com> .

Details of the confidentiality and pre-publicity policy may be found here <http://www.nature.com/authors/policies/confidentiality.html>

Privacy Policy | Update Profile